# Mixed-methods study of university students' perceptions of COVID-19 and media consumption from March 2020 –April 2022

**Madeleine Mant** [1]*, **Asal Aslemand**[2], **Andrew Prine**[3], **Alyson Holland**[4]

**1** Department of Anthropology, University of Toronto Mississauga, Mississauga, Ontario, Canada,
**2** Department of Mathematical & Computational Sciences, University of Toronto Mississauga, Mississauga, Ontario, Canada, **3** Groves Memorial Community Hospital, Fergus, Ontario, Canada, **4** Department of Family Medicine, McMaster University, Hamilton, Ontario, Canada

* maddy.mant@utoronto.ca

**Data Availability Statement:** Data from this study include sensitive personal medical information and potentially individually identifying demographic

## Abstract

Longitudinal mixed-methods research is necessary to understand the changing dynamics of pandemic perceptions, the adoption of health behaviours, and use of media during a public health emergency. During the first two years of the COVID-19 pandemic, we used a mixed-methods approach to survey young adults attending a large Canadian public university. Six online convenience survey samples were collected (Spring 2020, Summer 2020, Fall 2020, Spring 2021, Fall 2021, Spring 2022) (n = 4932) and 110 semi-structured interviews were conducted. Female gender was associated with higher perceptions of severity, susceptibility, and the adoption of new health behaviours. Perceptions of severity and anxiety/fear about contracting COVID-19 after reading/hearing a news report decreased overall over time, while perceptions of susceptibility increased through time overall. Social media was the most used form of media and was the form of media that participants judged to make them feel most anxious/fearful about contracting COVID-19. Those who felt anxious after hearing a COVID-19 news report were 8.43 times more likely to judge COVID-19 as severe and 2.07 times more likely to judge their own susceptibility as high. Interviews revealed perceptions of information overload, passive information intake, and a narrowed geographical focus over time. The decrease in judgements of COVID-19 severity with the accompanying increase in judgements of susceptibility over the first two years of the pandemic demonstrate the dynamics of changing pandemic attitudes. Health communication efforts targeting university students in future major health events need to consider these shifting dynamics and ensure that health information distributed via social media meets the needs of university students.

## Introduction

SARS-CoV-2 (COVID-19) was declared a pandemic by the World Health Organization on March 11, 2020. As of August 27, 2023, the ongoing COVID-19 pandemic has resulted in

information. Sharing these data in full would undermine the minimal risk ethical committee agreement and consent process. Participants did not provide consent for their transcripts and linked survey responses to be made public. However, de-identified data that support the results will be shared with researchers who (1) have approval from an institutional research ethics board for data use and (2) create a data use/sharing agreement with the University of Toronto. Researchers are welcome to apply for access from the corresponding author or the University of Toronto Human Research & Ethics Unit (HREU) Research Oversight & Compliance Office (ROCO) (contact via ethics.review@utoronto.ca).

**Funding:** This study was funded by the University of Toronto COVID-19 Action Initiative and by the University of Toronto Mississauga Department of Anthropology. The funders had no role in study design, data collection and analysis, decision to publish, or preparation of the manuscript.

**Competing interests:** The authors have declared that no competing interests exist.

nearly 7 million reported deaths and over 770 million confirmed cases [1]. Investigations into health promoting behaviours and the COVID-19 pandemic have uncovered differing responses to the pandemic attributed to factors such as gender, age, and income levels [2–4]. Survey studies regarding university student perceptions of severity, susceptibility, and media consumption arose in the early pandemic period, capturing moments in time, particularly in the spring and summer of 2020 [5–8].

Survey studies became an increasingly important tool to understand snapshot reactions to the COVID-19 pandemic as lockdowns occurred and educational institutions moved their operations online [9, 10]. Few studies, however, maintained a longitudinal lens on the rapidly changing COVID-19 situation, or incorporated mixed methods to understand the nuances of shifting perceptions by young adults. A mixed methods approach incorporating qualitative open-ended interviews can provide a nuanced understand of the beliefs and motivations that underlie the broader quantitative responses gathered through surveys. Responding to the over-whelming media coverage of the COVID-19 pandemic and the changing health mandates and recommendations over its first two years, the main objective of this research was to monitor university student perceptions of the ongoing pandemic and media use regarding COVID-19 longitudinally to understand if and how perceptions changed over time.

This research asked: (1) What demographic factors drive perceptions of severity, suscepti-bility, and health behaviour adoption over time?; (2) What media sources are being used by university students to access COVID-19-related information and what are students' percep-tions of their quality?; (3) How do perceptions of severity and susceptibility relate to media use regarding COVID-19 within the Health Belief Model (HBM)?

The HBM recognizes the relationship between health behaviours and pre-existing health beliefs [11–13]. This model predicts engagement in prevention activities by considering how beliefs about disease severity and individual susceptibility are understood by individuals in relation to existing cues-to-action (e.g., personal experience or education) and modifying fac-tors (e.g., age, gender, family history of disease). In the case of COVID-19, the HBM predicts that if a person considers the disease to be severe and themselves to be susceptible, then they are more likely to engage in prevention behaviors. These beliefs are influenced by cues-to-action, such as media, which can change perceptions of severity and susceptibility and there-fore change health behaviours. Previous studies, such as Oh and colleagues' investigation of adults in South Korea, have demonstrated that reading more about infectious disease out-breaks influences perceptions of disease severity [14]. Choi and colleagues, in a nationally rep-resentative online survey studying social media use during the 2015 Middle Eastern Respiratory Syndrome (MERS) outbreak in South Korea, found that participants who engaged with more MERS content on social media had higher risk perceptions scores [15].

Understanding university students' responses to large-scale disease events is important, as this demographic group has media habits and health/disease experiences that differ from other demographics. As noted in Mant et al. [6], young adults are often reported under the 'adult' (or 18+ years) category, limiting understanding of this unique group. This mixed-methods research used surveys and semi-structured interviews to investigate the relationship between demographic factors, perceptions of COVID-19 severity, perceptions of individual susceptibil-ity, time spent accessing COVID-19-related information from various forms of media, and perceptions of the reliability of media sources among university students during the first two years of the COVID-19 pandemic. This approach was selected to facilitate a holistic under-standing of the quantitative data, as perceptions were expected to change over time. Integrat-ing both quantitative and qualitative methods allows for a deeper understanding of university student perceptions of the COVID-19 pandemic, and an increased confidence in the results [16, 17]. To reach this demographic with public health updates regarding COVID-19 and

future pandemics, it is critical to have longitudinal understandings of how perceived severity and susceptibility–drivers within the health belief model–relate to demographic variables and the use of media.

## Methods

### Ethics statement

Ethics approval was obtained from the University of Toronto Research Ethics Board (#39169) and permission from the Provost and Vice-Provost to survey University of Toronto post-secondary students was granted on March 20, 2020. It was understood that some prospective participants would be under the age of 18; according to the *Tri-Council Policy Statement*: *Ethical Conduct for Research Involving Humans*, "in the case of post-secondary students recruited as research participants, the relevant criterion is not their age, but rather whether these students have the capacity to consent on their own behalf in the context of the particular study" [18]. This research project was deemed to have minimal risk and therefore any registered student could participate and consent on their own behalf. Survey participants indicated their consent to participate in the survey by clicking through to the second page of the survey and answering the questions. For interview participants, written informed consent was solicited from each participant before an interview was scheduled.

Recruitment for survey participation opened on March 20, 2020 and closed on April 22, 2022. Interviews were conducted between April 2, 2020 and April 9, 2022 (Fig 1).

### Survey recruitment

The six rounds of survey used a convenience sample of students at the University of Toronto. In each round, students were recruited through social media channels (Twitter, Facebook), a link on the University of Toronto Mississauga Department of Anthropology's website, and sharing the link with University of Toronto colleagues and student representatives. The first page of the survey contained a Letter of Information detailing the purpose of the study, data storage (all surveys to be assigned an anonymous ID number, files kept on a password-protected laptop, all data files to be password protected), and investigator contact information. Once a participant submitted the survey, they were provided with a Debrief Document outlining investigator contact information, health centre contact information, and crisis line contact details in case completing the survey was distressing. Compensation was offered through entry in a draw for a $50 gift card for participants who provided their email addresses.

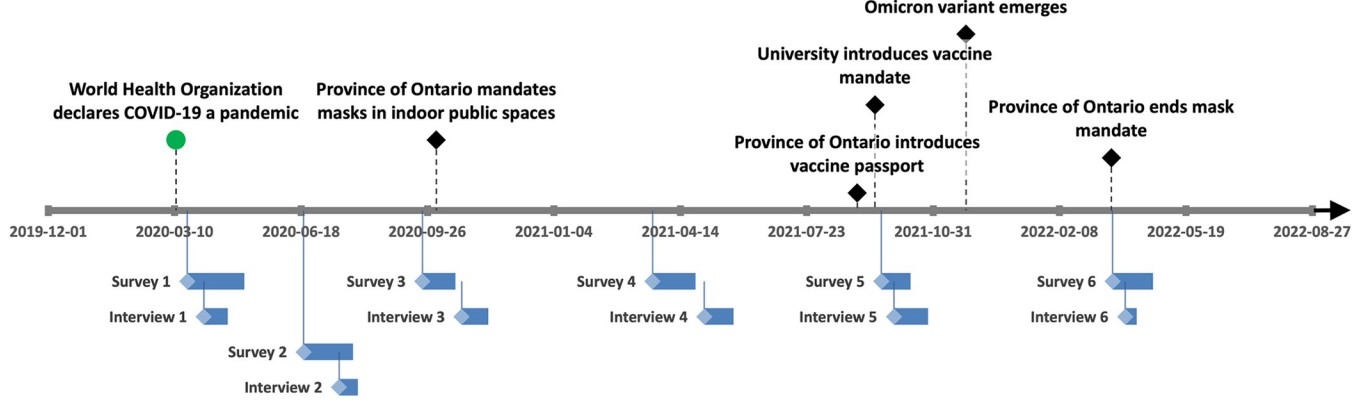

**Fig 1. Timeline of survey and interview periods with key contextual events.**

## Survey design and questions

The survey, hosted by Survey Planet, was adapted with permission from the authors by MM and AJH from a previous study surveying Canadian undergraduate students about Severe Acute Respiratory Syndrome (SARS) by Bergeron and Sanchez [19] who piloted and validated their survey. Participant demographics (i.e., age, gender identity, year and program of study at University of Toronto, location of residence, household income) were collected using multiple choice or free-form responses. Participants were asked to judge the severity of COVID-19 using a 7-point Likert scale from 1 ("Not severe") to 7 ("Very severe"). Judgements of personal susceptibility were gathered through free-form responses. Health behaviours (e.g., washing hands more, wearing a mask) were presented as a checklist and participants could select multiple choices. Participants were asked if they had regular access to the following media types: social media, internet news sources, radio, television, and magazines. Using adaptive questioning, if the participant answered "yes" to having access, they were asked to indicate in a typical week how many minutes per day they used the media source for accessing information about COVID-19. Participants were able to answer yes to multiple media sources. Participants were asked which media source caused them to feel the most fear and anxiety about contracting COVID-19. As designed and validated by Bergeron and Sanchez [19], the survey provided a 7-point Likert scale for participants to judge the success of each media source in bringing clear, concise, and unbiased information about the COVID-19 outbreak, measured from 1 ("Unsuccessful") to 7 ("Successful"); the scale was divided into Poor (1–3), Fair (4–5), and Good (6–7). Fear and anxiety concerning contracting COVID-19 after hearing or reading information about the outbreak was also scored on a 7-point Likert scale from 1 ("Not fearful/ anxious") to 7 ("Very fearful/anxious"). Each question appeared on a separate page and questions appeared in the same order for all survey participants. Participants were allowed to skip questions, answer "Prefer Not to Answer," or answer "Other" and enter an individual answer where necessary. Participants were not allowed to move backward in the survey to review or change their responses. The Checklist for Reporting Results of Internet E-Surveys (CHER-RIES) was used as a framework for reporting the results [20].

## Quantitative methodology

Survey responses for the six rounds of survey (Survey 1, Spring 2020 [March 20 –April 23], Survey 2, Summer 2020 [June 20 –July 28], Survey 3, Fall 2020 [September 22 –October 17], Survey 4, Spring 2021 [March 23 –April 25], Survey 5, Fall 2021 [September 20 –October 12], Survey 6, Spring 2022 [March 22 –April 22]) were analyzed using R statistical software, version 4.0.5. Any surveys with <75% of questions answered were removed. AH and MM coded and crosschecked the demographic survey data, and grouped the Likert scale questions for analysis. Logistic regressions were used to predict for young adults' judgement of severity of COVID-19; judgement of susceptibility to COVID-19; judgment of feeling anxious or fearful of acquiring COVID-19 after hearing a media report; perception of the media's ability to bring clear, concise, and unbiased COVID-19 information; and perception of the form of media that makes them most anxious and fearful about becoming infected with COVID-19. Poisson regression was used to predict for young adults' formation of new health behaviours, and the Chi-square test of independence was used to investigate the relationship between types of media used most for COVID-19 information and demographic factors. Participants' age, gender, program of study, and income level were treated as demographic factors. Age was treated as a continuous predictor variable; all remaining predictor variables were treated as categorical variables. P-values of $<0.05$ were considered statistically significant. The age range of participants selected for analysis was between 15 and 29 years (to align with the Statistics Canada

definition of youth). Two levels of gender identification were used for the analyses (male, female) for which there was a sufficient percentage of complete cases, with males as the reference category. Two levels of program of study were used for the analysis (health related, non-health related), with health related (e.g., health sciences, medicine) as a reference category. Three levels of income were used for the analysis (low, middle, high), with high level of income as a reference category. The time of survey data collection was used as a predictor variable, with Survey 1, Spring 2020 (Time 1) as a reference category. Table 1 shows the number of participants for each period of survey data collection. Since the student population of the University of Toronto is large (over 97,000) [21], the finite population correction [22] is ignored for evaluating whether this study's convenience sample size is representative. The required sample size is at least 385 cases to estimate population proportions within 5% margin of error with 95% confidence of the true proportion. The study's convenience sample sizes in each round of surveys meet this criterion. Following Riehm et al. [23:631], participants who reported "implausible values. . . (i.e., <0 minutes or >480 minutes [8 hours]" were removed from analysis.

## Interview participant recruitment

Participants were given the option at the end of the survey to provide their email address if they wanted to take part in an interview. We sought gender balance in each set of interviews, as most of the survey participants identified as female. All potential interview participants were emailed a Letter of Information that described the study objectives, participants' right to withdraw, and details of data storage and protection. Interviews were conducted by MM, AH, and AP over Skype or Zoom and lasted approximately one hour. A semi-structured interview guide (S1 Text) was used based upon the survey questions. Interview participants all received a $20 CAD gift card. The interviews took place during the following ranges: Spring 2020 (April 2–21); Summer 2020 (July 18 –August 1); Fall 2020 (October 23 –November 12); Spring 2021 (May 3–25); Fall 2021 (September 30 –October 26); Spring 2022 (April 1–9). In each round of interviews, the previous interviewees were contacted and asked if they wanted to participate again. These individuals were interviewed first and, using the principle of saturation to determine when a satisfactory number of interviews had been conducted, additional individuals were invited until it was determined that theme saturation had been achieved.

## Qualitative and integrative mixed-methods methodology

The benefit of a true mixed-methods approach was to use surveys to gather quantitative data on beliefs and perceptions that would then be measured longitudinally. Qualitative, open-ended interviews were then conducted to gather nuanced data in order to better contextualise the survey responses. This provides a wealth of rich data on the process by which our respondents generate their beliefs, opinions, and actions, while also highlighting areas of internal conflict. The survey and interview guides were designed with methodological triangulation in mind, wherein interviewees were asked about their perceptions of severity and susceptibility (key survey questions), as well as asked to describe their strategies using media to find information regarding COVID-19 (survey asked for estimates of media usage time). All interviews were transcribed verbatim and read multiple times by MM and AJH to create initial codes, guided by the research questions and thus, the survey questions (e.g., severity, susceptibility, time). Further codes were created inductively using the principles of qualitative content analysis described by Graneheim and Lundman [24]. NVivo12 was used to create code hierarchies and merge duplicate themes. All codes were discussed between three of the authors (MM, AJH, AP). Following the quantitative analysis, MM returned to the interview transcripts to

**Table 1. Demographic variables in all six survey rounds.**

| Variable | Categories | Survey 1 n (%) | Survey 2 n (%) | Survey 3 n (%) | Survey 4 n (%) | Survey 5 n (%) | Survey 6 n (%) |
|---|---|---|---|---|---|---|---|
| Age | 16–22 | 442 (80.8) | 331 (78.3) | 984 (84.6) | 807 (82.4) | 953 (85.7) | 583 (82.3) |
| | 23–29 | 105 (19.2) | 92 (21.7) | 179 (15.4) | 172 (17.6) | 159 (14.3) | 125 (17.7) |
| Gender | Female | 458 (83.7) | 341 (80.6) | 838 (72.1) | 748 (76.4) | 804 (72.3) | 528 (74.6) |
| | Male | 89 (16.3) | 82 (19.4) | 325 (27.9) | 231 (23.6) | 308 (27.7) | 180 (25.4) |
| Program of Study | Not health related | 487 (89.0) | 353 (83.5) | 1040 (89.4) | 829 (84.7) | 1039 (93.4) | 641 (90.5) |
| | Health related | 60 (11.0) | 70 (16.5) | 123 (10.6) | 149 (15.2) | 73 (6.6) | 67 (9.5) |
| Household income | Low (<$24,999-$74,999) | 305 (55.8) | 239 (56.5) | 617 (53.1) | 524 (53.5) | 588 (52.9) | 360 (50.8) |
| | Middle ($75,000 - $149,999) | 190 (34.7) | 134 (31.7) | 425 (36.5) | 333 (34.0) | 383 (34.4) | 251 (35.5) |
| | High ($150,000 +) | 52 (9.5) | 50 (11.8) | 121 (10.4) | 122 (12.5) | 141 (12.7) | 97 (13.7) |
| Total participants included for demographic analysis | | 547 | 423 | 1163 | 979 | 1112 | 708 |

Descriptive statistics are calculated whereby n = number of participants in each category in each survey. Different aspects of Surveys 1, 2, and 3 were previously analyzed in Mant et al. [6, 25]. The total number of participants in the present study differs due to the age selection criteria outlined above.

determine if contextual details existed to understand the changes in perceptions and focus over time.

## Results

Survey responses were analyzed for the six iterations: Survey 1, Spring 2020 (n = 592); Survey 2, Summer 2020 (n = 483); Survey 3, Fall 2020 (n = 1269); Survey 4, Spring 2021 (n = 1066); Survey 5, Fall 2021 (n = 1241); and Survey 6, Spring 2022 (n = 793). Results from Survey 1 regarding perceptions of susceptibility and severity, and the adoption of new health behaviours appear in Mant et al. [6]. The demographics of the Survey 2 and Survey 3 participants and their perceptions of severity have previously appeared in Mant et al. [25], in this context as predictor variables in binary and multinominal logistic regression analyses investigating likelihood of vaccine uptake. Of the 5124 total surveys in the cleaned sample, 4932 responses included responses to all demographic categories and aligned with the Statistics Canada definition of youth (15–29) and were thus included in the analyses (Table 1). The majority of the survey participants (83%) were between 19 to 22 years old, with a mean of 20.5 years. Approximately 75% of participants identified as female and about 25% as male. There were a number of individuals who self-identified as gender non-conforming, non-binary, selected to self-describe, or elected not to indicate their gender. These categories were too small to be included in the statistical models; including individuals who self-identified within a gender variant category was thus made a priority for the interviews. The majority of participants, about 89%, were enrolled in non-health related programs. Approximately 53% of the participants were from low-income families, 35% were from middle-income families, and 12% were from high-income families.

### Participants' demographic characteristics and judgement of severity of COVID-19, judgement of susceptibility to COVID-19, and adoption of new health behaviours over time

The response variable, participants' judgement of severity of COVID-19, was grouped as low (1–3) and high (4–7), with low as a reference category (Table 2). Likelihood ratio tests in the binary logistic regression analysis indicate that gender ($p < 0.001$) and time of survey data collection ($p < 0.001$) predicted judgement of the severity of COVID-19 (Table A in S2 Text).

**Table 2. Survey results for judgement of severity of COVID-19, judgement of personal susceptibility, and feelings of anxiety/fear after hearing/reading a news report regarding COVID-19.**

| Variable | Category | Survey 1 n (%) | Survey 2 n (%) | Survey 3 n (%) | Survey 4 n (%) | Survey 5 n (%) | Survey 6 n (%) | Total n (%) |
|---|---|---|---|---|---|---|---|---|
| Judgment of severity of COVID | Low (1–3) | 18 (3.3) | 25 (6.0) | 77 (6.7) | 70 (7.2) | 99 (8.9) | 128 (18.1) | 417 (8.5) |
| | High (4–7) | 526 (96.7) | 395 (94.0) | 1075 (93.3) | 903 (92.8) | 1008 (91.1) | 581 (81.9) | 4488 (91.5) |
| Judgement of susceptibility | Yes | 309 (57.3) | 254 (60.3) | 685 (59.0) | 578 (59.1) | 562 (50.6) | 449 (63.4) | 2837 (57.7) |
| | No | 230 (42.7) | 167 (39.7) | 477 (41.0) | 400 (40.9) | 548 (49.4) | 259 (36.6) | 2081 (42.3) |
| Anxiety/fear after COVID-19 report | Low (1–3) | 120 (22.4) | 118 (28.2) | 290 (25.0) | 201 (20.6) | 302 (27.3) | 250 (35.3) | 1281 (26.1) |
| | High (4–7) | 415 (77.6) | 300 (71.8) | 868 (75.0) | 774 (79.4) | 804 (72.7) | 459 (64.7) | 3620 (73.9) |

Controlling for all other factors, female students were 2.55 times more likely to judge the severity of COVID-19 as high (Table B in S2 Text). Overall, predicted probabilities of judging severity as high decreased over time (Fig 2).

The response variable–participants' judgements of their susceptibility to COVID-19, was grouped as yes (susceptible) and no (not susceptible), with no as a reference category (Table 2). Likelihood ratio tests in the binary logistic regression analysis (Table A in S2 Text) indicate that age ($p < 0.001$), gender ($p < 0.001$), and time of survey data collection ($p < 0.001$) were related to judgements of susceptibility. Allowing for all other factors, with each additional one-year increase in age, the odds of perceiving oneself to be susceptible to

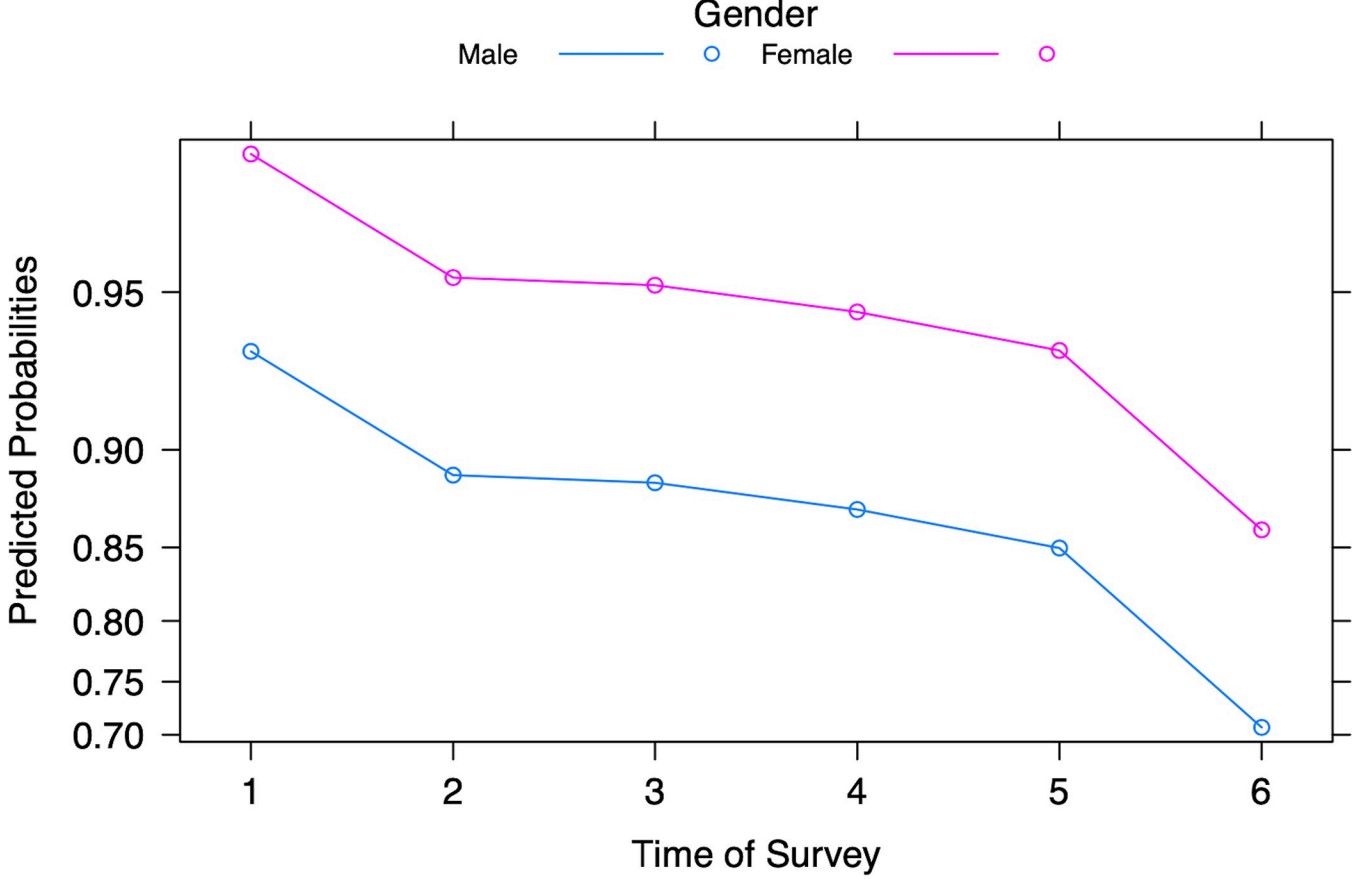

**Fig 2. Predicted probabilities of judging severity of COVID-19 to be high for male and female participants for all surveys.**

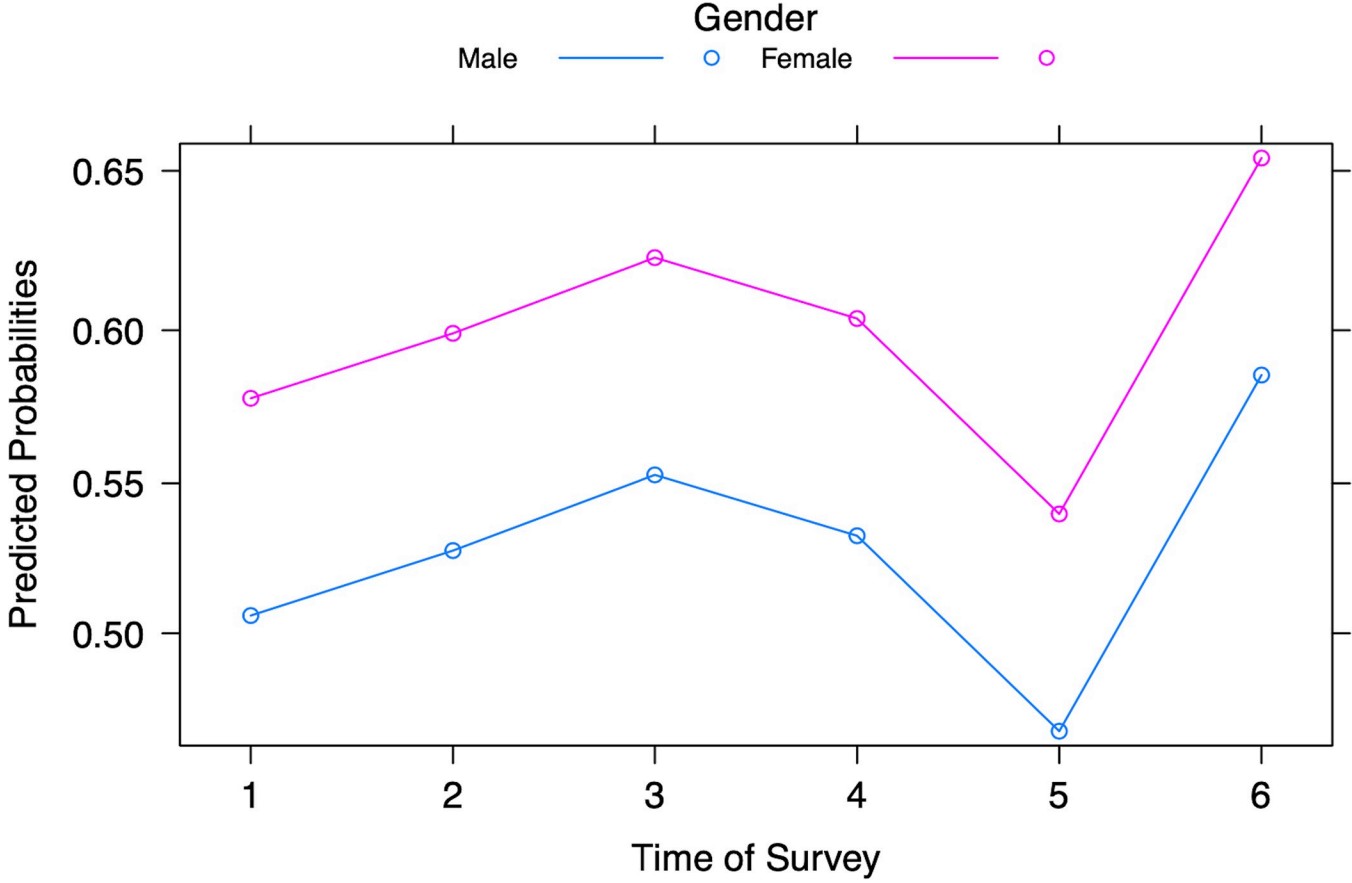

**Fig 3. Predicted probabilities of judgements of susceptibility to COVID-19 for male and female participants from Surveys 1 to 6.**

COVID-19 increased by 17%. Female students were 1.34x more likely to perceive themselves to be susceptible to COVID-19 and compared to those surveyed during Spring 2020 (Survey 1), participants in Spring 2022 (Survey 6) were 1.38x more likely to perceive themselves as susceptible to catching COVID-19 (Table B in S2 Text). The pattern of susceptibility changes over time, with the lowest predicted probability found in Survey 5 (Fall 2021) and the highest in Survey 6 (Spring 2022) (Fig 3).

New health behaviours were treated as quantitative response variables; these were: changing travel plans, wearing a mask, washing hands more often, using hand sanitizer, social distancing, cleaning more, buying extra food/supplies, seeking a doctor's help/advice, seeking hospital help/advice, calling public health, being tested for COVID-19, or choosing to pre-emptively get a COVID-19 test. The likelihood ratio tests in the Poisson regression analysis (Table A in S2 Text) indicates that factors predicting new health behaviours included gender ($p < 0.001$), and time of survey data collection ($p < 0.001$). Allowing for all other factors, compared to the male students, female students were about 1.2x more likely to indicate new health behaviours (Table B in S2 Text). Controlling for all other variables, compared to those surveyed in Spring 2020 (Survey 1), the participants surveyed in Surveys 2 to 5 were about 1.2x more likely to indicate new health behaviours. Fig 4 displays the number of health behaviours indicated by male and female participants in all surveys. The overall number of new health behaviours increases from Surveys 1 to 3 and then decreases from Surveys 4 to 6.

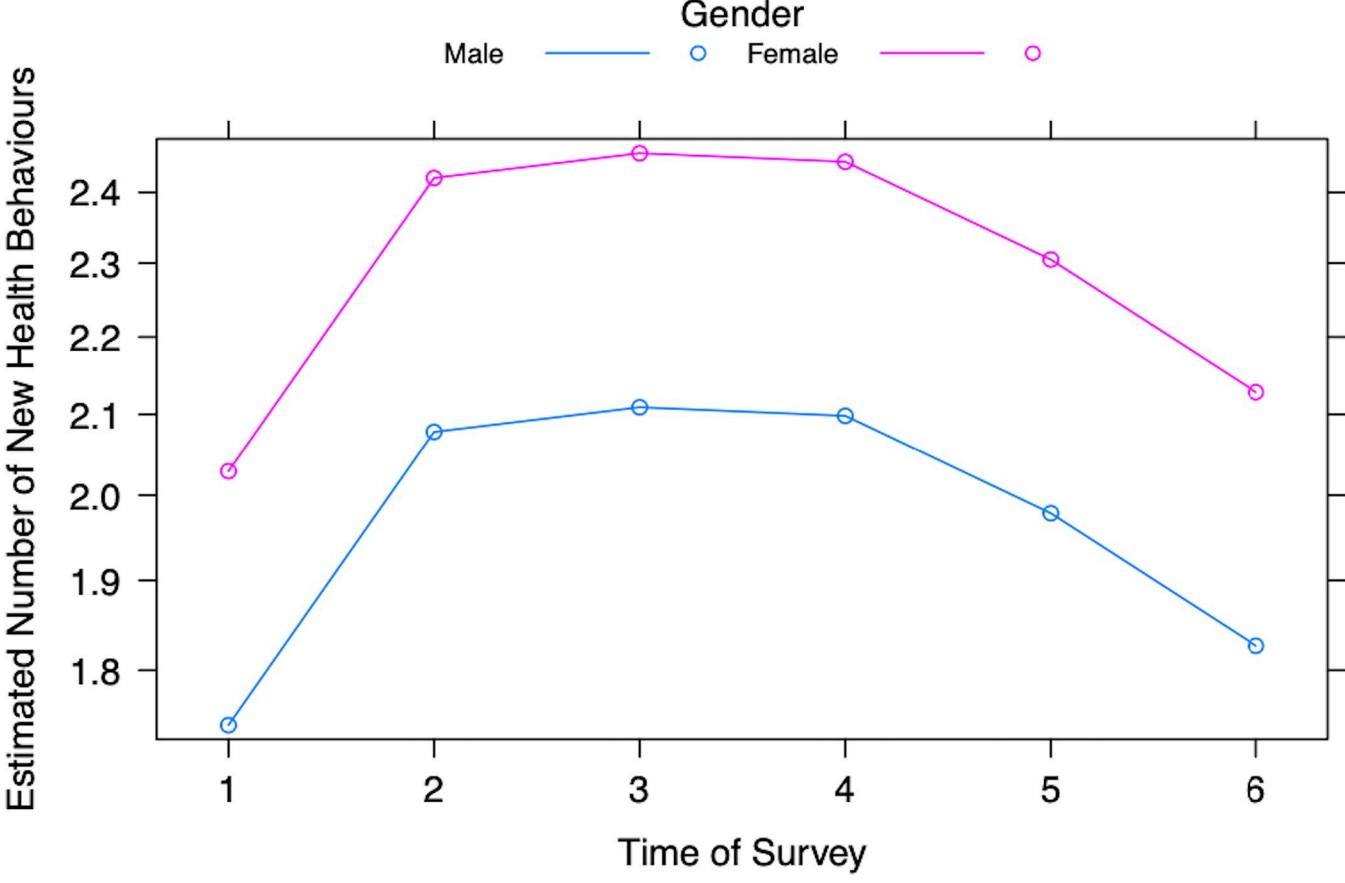

**Fig 4. Estimated number of new health behaviours adopted by male and female participants from Surveys 1–6.**

Chi-square tests of independence were carried out to investigate the relationship between each health behaviour and the time of data collection. The health behaviours found to have a significant relationship to time of data collection are presented in Table 3, along with the number of participants indicating each behaviour (participants were able to select multiple

**Table 3. Chi-square tests of independence for relationships between health behaviours and time of data collection, number of individuals who indicated health behaviour in Surveys 1–6.**

| Health behaviour | $\chi^2$ | df | p-value | Survey 1 n (%) | Survey 2 n (%) | Survey 3 n (%) | Survey 4 n (%) | Survey 5 n (%) | Survey 6 n (%) |
|---|---|---|---|---|---|---|---|---|---|
| Changed travel plans | 314.99 | 5 | < 0.001 | 497 (90.9) | 283 (66.9) | 695 (59.8) | 599 (61.2) | 587 (52.8) | 321 (45.3) |
| Wear a mask | 1763.59 | 5 | < 0.001 | 115 (21.0) | 355 (83.9) | 1079 (92.8) | 917 (93.7) | 1015 (91.3) | 638 (90.1) |
| Wash hands more | 78.36 | 5 | < 0.001 | 472 (86.3) | 358 (84.6) | 972 (83.6) | 978 (99.9) | 860 (77.3) | 497 (70.2) |
| Use hand sanitizer | 72.36 | 5 | < 0.001 | 376 (68.7) | 336 (79.4) | 976 (83.9) | 780 (79.7) | 904 (81.3) | 514 (72.6) |
| Social distancing | 252.93 | 5 | < 0.001 | 500 (91.4) | 379 (89.6) | 1019 (87.6) | 875 (89.4) | 873 (78.5) | 467 (66.0) |
| Self-Isolating | 528.94 | 5 | < 0.001 | 0 | 199 (47.0) | 456 (39.2) | 434 (44.3) | 214 (19.2) | 149 (21.0) |
| Cleaning more | 467.83 | 5 | < 0.001 | 0 | 210 (49.6) | 587 (50.5) | 474 (48.4) | 488 (43.9) | 260 (36.7) |
| Buy extra food/supplies | 287.21 | 5 | < 0.001 | 310 (56.7) | 157 (37.1) | 373 (32.1) | 296 (30.2) | 227 (20.4) | 133 (18.8) |
| Tested for COVID-19 | 573.65 | 5 | < 0.001 | 0 | 23 (5.4) | 155 (13.3) | 281 (28.7) | 375 (33.7) | 318 (44.9) |
| Choose to get a COVID-19 test | 581.74 | 5 | < 0.001 | 0 | 0 | 140 (12.0) | 225 (23.0) | 321 (28.9) | 305 (43.1) |
| Total survey participants in sample | | | | 547 | 423 | 1163 | 979 | 1112 | 708 |

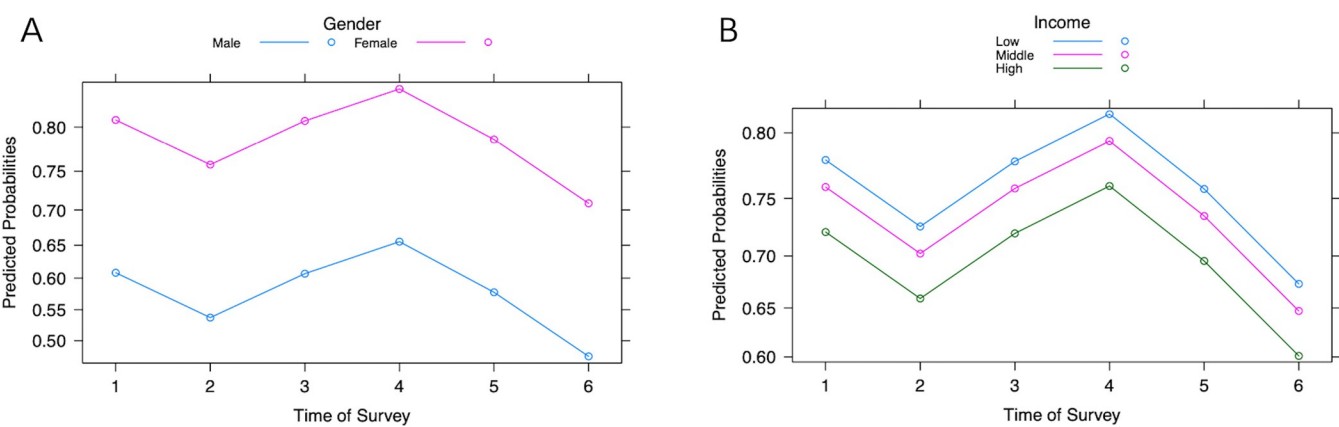

**Fig 5.** Predicted probabilities of feeling anxiety/fear about acquiring COVID-19 after reading/hearing a news report for (A) gender and (B) income categories.

options). For example, the number of participants indicating that they had changed travel plans in response to COVID-19 decreased from Survey 1 (90.7%) to Survey 6 (45.3%), while wearing a mask increased sharply between Surveys 1 (21.0%) and 2 (83.9%) and remained high (Survey 6 = 89.9%). Participants were less likely to social distance and wash their hands over time. Buying extra food/supplies decreased over time, while the number of participants choosing to get or being administered a COVID-19 test increased over time.

## Judgement of feelings of anxiety/fear of acquiring COVID-19 after reading/hearing a news report about COVID

Participants' judgement of their feelings of anxiety/fear of acquiring COVID-19 after reading/hearing a news report about COVID were grouped as low (1–3) and high (4–7) (Table 2) and used as the response variable. Likelihood ratio tests in the binary logistic regression analysis indicate that gender ($p < 0.001$), income ($p < 0.05$), and time of survey data collection ($p < 0.001$) were predictive of an individual's judgement of anxiety/fear of contracting COVID-19 (Table C in S2 Text). Controlling for all other factors, female students were 2.7x more likely than male students to feel anxious/fearful of catching COVID-19 after reading/hearing a news report about COVID, and those from low-income families were 1.37x more likely to feel anxiety/fear than those from high-income families (Table D in S2 Text). Controlling for all other variables, compared to those surveyed in Survey 1 (Spring 2020), the cohort in Survey 6 (Spring 2022) had 42% lower odds of feeling anxiety/fear about catching COVID-19 after reading/hearing a COVID-19 news report (Table D in S2 Text). Fig 5 displays the predicted probabilities of participants feeling anxiety/fear of acquiring COVID-19 after hearing/reading a news report for (A) male and female participants and (B) participants with different income levels. Female participants' rated anxiety/fear was consistently higher than male participants; low-income participants rated anxiety/fear consistently higher than those in other income brackets.

## Time spent using media sources and perceptions of media sources

Participants were asked which media sources they had access to and how much time (in minutes) they spent accessing these media sources for COVID-19 related information daily. Table E in S2 Text displays the minimum, maximum, median, average, and interquartile range of time spent using each media source for COVID-19 information. Reported median time spent using each type of media decreased for all types of media from Survey 1

to Survey 6. Social media was used most to access COVID-19 information, followed by internet news.

Participants were asked to judge the ability of each media source to present clear, concise, and unbiased information about COVID-19. Chi-square tests of independence indicate that there are statistically significant associations between perceptions of the media sources and the time of survey (Table 4). From Spring 2020 (Survey 1) to Fall 2020 (Survey 3), most participants judged social media to be doing a middling job, whereas from Spring 2021 (Survey 4) to Spring 2022 (Survey 6), the majority judged social media to be doing a poor job. The remainder of the media sources were judged consistently by the majority of participants to be doing a middling job in all surveys.

Participants were asked which form of media made them feel the most anxiety/fear about becoming infected with COVID-19 (Table 5). Overall, 44.5% of participants indicated that social media made them feel the most anxiety/fear; 26.7% indicated it was internet news and 23.6% cited television. The form of media making participants feel anxiety/fear (i.e., social media, internet, TV, other) was the response variable, with 'other' as the reference category. The likelihood ratios tests in the multinomial logistic regression analysis (Table 6) indicates that age, gender, and program of study ($p < 0.001$) were all significant predictors. Allowing for all other predictors, with each additional one-year increase in age, the odds of perceiving social media as a source of anxiety/fear decreased by about 11%, internet by about 11%, and television by 13% (Table E in S2 Text). Controlling for all other factors, female participants were about 1.9x more likely to note that social media and internet and 2x more likely to note television was the main form of media making them feel anxious/fearful. Those in non-health related programs were 55% less likely to feel anxiety/fear regarding social media, 62% less likely to feel anxiety/fear related to internet, and 59% less likely to feel anxiety/fear regarding television, though these results should be interpreted with caution since 89% of survey participants were enrolled in non-health related programs.

## Relationships between judgements of severity, anxiety/fear, and media use

We asked what relationship existed between participants' judgements of the severity of COVID-19, time spent using media to look at reports related to COVID-19, judgements of anxiety/fear of catching COVID-19 after reading/hearing a COVID-19 news report, and the type of media that makes them feel anxiety/fear. The response variable, judgment of COVID-19 severity, was grouped as low (1–3) and high (4–7), with low as a reference category. Quantitative predictor variables included participants' reported times spent looking at COVID-19-related media (Table 7). The categorical predictor variables included participants' reported level of anxiety/fear about catching COVID-19 after reading/hearing a COVID-19-related news report, which was grouped as low (1–3) and high (4–7), with low as a reference category, and type of media that contributes most to feelings of anxiety/fear. Very few individuals selected Magazine, All, None, and Other, thus they were grouped together to make reasonable comparisons between media type categories. Likelihood ratio tests in the binary logistic regression analysis (Table 8) indicates that feeling anxiety/fear about catching COVID-19 after reading a COVID-19 news report was a predictor of judgements of the severity of COVID-19. The results of the binary logistic regression analysis (Table F in S2 Text) indicates that, controlling for type of media, individuals feeling anxiety/fear after reading a COVID-19 news report were 8.43x more likely to judge the severity of COVID-19 to be high. Allowing for the variable of feeling anxiety/fear after reading/hearing a COVID-19 news report, the odds of judging the severity of COVID-19 to be high was 1.51x higher for those who reported that social media made them feel the most anxiety/fear. There were no statistically significant associations

**Table 4. Association between perceptions of media in bringing clear, concise, and unbiased information about COVID-19 and time of survey with summary statistics.**

| Relationship with time of survey | $\chi^2$ | df | p-value |
|---|---|---|---|
| Social media[a] | 99.36 | 10 | < 0.001 |
| Internet[a] | 77.29 | 10 | < 0.001 |
| Radio[a] | 53.26 | 10 | < 0.001 |
| TV[a] | 50.79 | 10 | < 0.001 |
| Magazine/other | 5.74 | 10 | 0.220 |

| Social Media | | | | |
|---|---|---|---|---|
| Survey | Poor | Middling | Successful | Total |
| 1 | 212 (39.8%) | 274 (51.5%) | 46 (8.6%) | 532 |
| 2 | 153 (36.6%) | 224 (53.6%) | 41 (9.8%) | 418 |
| 3 | 408 (37.0%) | 530 (48.0%) | 166 (15.0%) | 1104 |
| 4 | 428 (47.0%) | 400 (44.0%) | 82 (9.0%) | 910 |
| 5 | 513 (50.3%) | 427 (41.9%) | 79 (7.8%) | 1019 |
| 6 | 335 (51.5%) | 273 (42.0%) | 42 (6.5%) | 650 |
| Total | 2049 (44.2%) | 2128 (45.9%) | 456 (9.8%) | 4633 |

| Internet | | | | |
|---|---|---|---|---|
| Survey | Poor | Middling | Successful | Total |
| 1 | 64 (13.2%) | 277 (57.1%) | 144 (29.7%) | 485 |
| 2 | 74 (18.0%) | 238 (58.0%) | 98 (23.9%) | 410 |
| 3 | 188 (18.8%) | 527 (52.8%) | 284 (28.4%) | 999 |
| 4 | 190 (22.6%) | 479 (56.9%) | 173 (20.5%) | 842 |
| 5 | 213 (24.0%) | 499 (56.3%) | 175 (19.7%) | 887 |
| 6 | 157 (29.3%) | 284 (53.0%) | 95 (17.7%) | 536 |
| Total | 886 (21.3%) | 2304 (55.4%) | 969 (23.3%) | 4159 |

| Radio | | | | |
|---|---|---|---|---|
| Survey | Poor | Middling | Successful | Total |
| 1 | 13 (13.3%) | 54 (55.1%) | 31 (31.6%) | 98 |
| 2 | 56 (25.9%) | 128 (59.3%) | 32 (14.8%) | 216 |
| 3 | 114 (26.1%) | 267 (61.1%) | 56 (12.8%) | 437 |
| 4 | 77 (22.7%) | 218 (64.3%) | 44 (13.0%) | 339 |
| 5 | 79 (24.5%) | 213 (66.1%) | 30 (9.3%) | 322 |
| 6 | 81 (35.4%) | 129 (56.3%) | 19 (8.3%) | 229 |
| Total | 420 (25.6%) | 1009 (61.5%) | 212 (12.9%) | 1641 |

| Television | | | | |
|---|---|---|---|---|
| Survey | Poor | Middling | Successful | Total |
| 1 | 32 (11.7%) | 137 (50.2%) | 104 (38.1%) | 273 |
| 2 | 99 (25.2%) | 217 (55.2%) | 77 (19.6%) | 393 |
| 3 | 133 (21.4%) | 309 (49.8%) | 179 (28.8%) | 621 |
| 4 | 105 (19.3%) | 307 (56.5%) | 131 (24.1%) | 543 |
| 5 | 97 (19.7%) | 268 (54.5%) | 127 (25.8%) | 492 |
| 6 | 81 (26.0%) | 166 (53.4%) | 64 (20.6%) | 311 |
| Total | 547 (20.8%) | 1404 (53.3%) | 682 (25.9%) | 2633 |

| Magazines/Other | | | | |
|---|---|---|---|---|
| Survey | Poor | Middling | Successful | Total |
| 1 | 5 (35.7%) | 7 (50.0%) | 2 (14.3%) | 14 |
| 2 | 22 (46.8%) | 23 (48.9%) | 2 (4.3%) | 47 |
| 3 | 34 (40.5%) | 41 (48.8%) | 9 (10.7%) | 84 |

*(Continued)*

**Table 4.** (Continued)

| Relationship with time of survey | $\chi^2$ | df | p-value | |
|---|---|---|---|---|
| 4 | 20 (32.8%) | 35 (57.4%) | 6 (9.8%) | 61 |
| 5 | 31 (50.0%) | 29 (46.8%) | 2 (3.2%) | 62 |
| 6 | 22 (61.1%) | 14 (38.9%) | 0 (0.0%) | 36 |
| Total | 134 (44.1%) | 149 (49.0%) | 21 (6.9%) | 304 |

a. $p < 0.05$.

**Table 5. Participants' judgements of which media source made them feel the most anxiety/fear about becoming infected with COVID-19.**

| | Survey 1 n (%) | Survey 2 n (%) | Survey 3 n (%) | Survey 4 n (%) | Survey 5 n (%) | Survey 6 n (%) | Total n (%) |
|---|---|---|---|---|---|---|---|
| Social media | 253 (46.8) | 193 (45.8) | 484 (41.8) | 431 (44.7) | 481 (43.9) | 328 (46.7) | 2170 (44.5) |
| Internet | 131 (24.2) | 105 (24.9) | 332 (28.7) | 244 (25.3) | 305 (27.8) | 187 (26.6) | 1304 (26.7) |
| Radio | 4 (0.7) | 2 (0.5) | 8 (0.7) | 7 (0.7) | 7 (0.6) | 3 (0.4) | 31 (0.6) |
| TV | 138 (25.5) | 104 (24.7) | 278 (24.0) | 230 (23.9) | 253 (23.1) | 148 (21.1) | 1151 (23.6) |
| Magazines | 0 | 1 (0.2) | 0 | 1 (0.1) | 2 (0.2) | 1 (0.1) | 5 (0.1) |
| All | 0 | 4 (1.0) | 2 (0.2) | 5 (0.5) | 5 (0.5) | 1 (0.1) | 17 (0.3) |
| None | 3 (0.6) | 2 (0.5) | 13 (1.1) | 29 (3.0) | 31 (2.8) | 20 (2.8) | 98 (2.0) |
| Other | 12 (2.2) | 10 (2.4) | 40 (3.5) | 17 (1.8) | 12 (1.1) | 14 (2.0) | 105 (2.2) |
| Total | 541 | 421 | 1157 | 964 | 1096 | 702 | 4881 |

between anxiety/fear after reading/hearing a COVID-19 news report and time spent using media.

## Relationships between judgements of susceptibility, anxiety/fear, and media use

Similarly, we investigated the relationship between participants' judgement of their susceptibility to catching COVID-19 (response variable groups as yes [susceptible] and no [not susceptible]), with no as the reference category. The quantitative and categorical predictor variables were the same as above. The likelihood ratio tests in the binary logistic regression analysis (Table 8) indicates that an individual's judgement of their own susceptibility to COVID-19 was predicted by their indication of feeling anxiety/fear after reading/hearing a COVID-19

**Table 6. Main effects of predictors on forms of media that make participants feel most anxiety/fear about becoming infected with COVID-19.**

| Predictor | $\chi^2$ | df | p-value |
|---|---|---|---|
| Age[a] | 23.1120 | 3 | < 0.001 |
| Gender[a] | 22.4630 | 3 | < 0.001 |
| Program[a] | 14.8341 | 3 | 0.002 |
| Income | 6.3397 | 6 | 0.386 |
| Time | 17.2293 | 15 | 0.305 |

a. $p < 0.05$.

**Table 7. Participants' report time (in minutes) using each media source per day from Surveys 1–6.**

|  | Survey 1 | Survey 2 | Survey 3 | Survey 4 | Survey 5 | Survey 6 |
|---|---|---|---|---|---|---|
| **n** | **543** | **420** | **1162** | **976** | **1111** | **708** |
| | | | *Social Media* | | | |
| Minimum | 0 | 0 | 0 | 0 | 0 | 0 |
| Q1 | 10 | 10 | 5 | 0 | 0 | 0 |
| Median | 30 | 20 | 10 | 10 | 8 | 5 |
| Q3 | 60 | 30 | 30 | 20 | 20 | 15 |
| Maximum | 800 | 900 | 800 | 900 | 900 | 700 |
| | | | *Internet* | | | |
| Minimum | 0 | 0 | 0 | 0 | 0 | 0 |
| Q1 | 9 | 5 | 0 | 0 | 0 | 0 |
| Median | 20 | 10 | 10 | 5 | 5 | 3 |
| Q3 | 40 | 30 | 20 | 20 | 15 | 10 |
| Maximum | 840 | 500 | 900 | 800 | 840 | 420 |
| | | | *Radio* | | | |
| Minimum | 0 | 0 | 0 | 0 | 0 | 0 |
| Q1 | 0 | 0 | 0 | 0 | 0 | 0 |
| Median | 0 | 0 | 0 | 0 | 0 | 0 |
| Q3 | 0 | 0 | 0 | 0 | 0 | 0 |
| Maximum | 500 | 500 | 150 | 200 | 90 | 300 |
| | | | *Television* | | | |
| Minimum | 0 | 0 | 0 | 0 | 0 | 0 |
| Q1 | 0 | 0 | 0 | 0 | 0 | 0 |
| Median | 0 | 2 | 0 | 0 | 0 | 0 |
| Q3 | 30 | 20 | 10 | 10 | 9 | 5 |
| Maximum | 840 | 500 | 800 | 720 | 700 | 840 |
| | | | *Magazines* | | | |
| Minimum | 0 | 0 | 0 | 0 | 0 | 0 |
| Q1 | 0 | 0 | 0 | 0 | 0 | 0 |
| Median | 0 | 0 | 0 | 0 | 0 | 0 |
| Q3 | 0 | 0 | 0 | 0 | 0 | 0 |
| Maximum | 200 | 200 | 90 | 57 | 150 | 100 |

news report ($p < 0.001$). Controlling for the type of media, participants who felt anxiety/fear after reading/hearing a COVID-19 media report were 2.07x more likely to indicate that they were susceptible to COVID-19 (Table F in S2 Text).

**Table 8. Main effects of predictors on the judgement of severity of and susceptibility to COVID-19.**

| Predictor | $\chi^2$ | df | p-value |
|---|---|---|---|
| | *Severity of COVID-19* | | |
| Fear of acquiring COVID[a] | 385.71 | 1 | < 0.001 |
| Type of media | 5.27 | 3 | 0.153 |
| | *Susceptibility to COVID-19* | | |
| Fear of acquiring COVID[a] | 122.28 | 1 | < 0.001 |
| Type of media | 2.75 | 3 | 0.432 |

Frequency data are available in Tables 2 and 5.

a. $p < 0.05$.

**Table 9. Interview themes and topics.**

| Themes | Categories | Survey 1 (Spring 2020) | Survey 2 (Summer 2020) | Survey 3 (Fall 2020) | Survey 4 (Spring 2021) | Survey 5 (Fall 2021) | Survey 6 (Spring 2022) |
|---|---|---|---|---|---|---|---|
| *Trust* | Expressed confidence about media source(s) | X* | X | X | X | X | X |
| | Expressed doubt about media source(s) | X | X | X | X | X | X |
| | Strategies for checking reliability of source(s) | X | X | | | | |
| *Time* | Actively searching for COVID-19 information | X | X | X | Only for vaccine information | Only for mandates/ rules | Only for mandates/ rules |
| | Passively taking in COVID-19 information | X | X | X | X | X | X |
| | Less than 'before' | | X | X | X Exception: vaccine booking | X | X |
| *Stress/Mood* | Effect of overwhelming/ negative media coverage | X | X | X | X | X | |
| *Family* | Family as source of COVID-19 information | X | X | | | X | X |
| | Keeping track of COVID-19 information for family | | | | X | | |
| *Geographical focus* | Local | X | X | X | X | X | X |
| | Provincial | X | X | X | X | X | |
| | National (Canada) | X | X | X | X | Only for travel restrictions | Only for vaccine uptake |
| | International (where family/ friends are) | X | X | X | X | Only for travel restrictions | Only for travel restrictions |
| | International (for case count/ mandate comparisons) | X | X | X | | | |
| *Topics* | Newly introduced topics | Case counts, school closure | Conspiracy theories, masks, vaccine trials, bubbles/pods | Online school, lockdown | Booking vaccines, variants, situation in India and Brazil | Vaccine passports, vaccine mandates, anti-vaxx movement | Anti-vaxx movement, Omicron, invasion of Ukraine |
| | Continued topics | | Case counts | Case counts, masks, vaccine trials | Case counts | Case counts | Case counts, vaccine mandates |

* = presence of theme

## Qualitative results from interviews regarding media usage

Interviews accompanied all six rounds of the survey; interviewees between the ages of 18 to 29 were considered for this analysis (Table G in S2 Text). Interviews included questions regarding media usage (S1 Text). The six sets of interviews expanded upon the themes identified in Table 9: trust, time, stress/mood, and family. Supporting quotations for the identified themes are displayed in Table 10.

**Trust.** *Expressed confidence about media source(s).* In all rounds of interviews, students identified examples of which media/news sources they found trustworthy and/or reliable.

*Expressed doubt about media source(s).* Similarly, in all rounds of interviews, students highlighted which media sources they used that they questioned regarding possible bias or unreliability.

*Strategies for checking reliability of source(s).* In the Spring 2020 (Survey 1) and Summer 2020 (Survey 2) interviews, students outlined strategies they used for crosschecking the

reliability or accuracy of the media sources they were using to access information about COVID-19. These included strategies such as "*I always go to the comments just to see what if it's reliable and just to see where it came from*"; "*I will also look for where they got their stats and information from*"; "*if I see an article, I will probably look at where they were funded*"; and "*I always go to Google Scholar and like find something about it to confirm that it's right.*" This theme was not present in the subsequent interviews.

**Time.**   *Actively searching for COVID-19 information.* In the three rounds of interviews in 2020, students expressed that they were actively searching for information about COVID-19. Examples included using Google and government or university websites to look up information about the emerging disease, symptoms, and case counts. In Spring 2021 (Survey 4) there was a tonal change and students expressed that they were only looking up information that related to vaccines (i.e., outcomes of trials, booking vaccine appointments). In both Fall 2021 (Survey 5) and Spring 2022 (Survey 6), the interviewees noted that they were only actively

**Table 10. Interview participant quotations illustrating themes.**

| Themes | | Illustrative interview quotations |
|---|---|---|
| *Trust* | Expressed confidence about media source(s) | The news outlets like CBC or CTV, they're very accurate, because there's definitely a delay in the um information transfer that they get and then they have to filter a lot so they don't like panic the public (Spring 2020, Survey 1).<br>But overall, it's the WHO, CDC, like any company, for lack of a better word, that is reputable in terms of medicine and stuff like that (Spring 2020, Survey 1).<br>I trust my local news station I think (Summer 2020, Survey 2).<br>I believe the statistics and working with what the doctors are saying in terms of the news sources, in terms of people's personal experiences. (Summer 2020, Survey 2).<br>When I do like research for school I look- go deep down into like peer reviewed, you know all that but, like I guess, you know, as of right now I'm just trusting the government website and the news as much as I can (Spring 2021, Survey 4). |
| | Expressed doubt about media source(s) | I know that on Reddit, because anybody can like post anything, you kind of have to be a little more careful about like, is that information true or not (Spring 2020, Survey 1).<br>I see it when like I use like use social media normally, but I mostly don't trust what I see there (Spring 2020, Survey 1).<br>I am always hesitant to trust things on social media because it's so easy to post opinions as opposed to facts. I would say I am confident people are posting some information that is true, but I don't necessarily trust any post as reliable (Spring 2020, Survey 1).<br>So like for basic information I think it's good, but in terms of like caring about what politicians have to say and stuff I think social media is definitely not the best because there is a lot of bias (Fall 2021, Survey 5). |
| | Strategies for checking reliability of source(s) | If I see an Instagram or Facebook post I always go to the comments just to see what if it's reliable and just to see where it came from (Spring 2020, Survey 1).<br>I feel like a lot of the information I've gotten has been valid because when I Google it or search it, I'm like oh this actually holds up (Spring 2020, Survey 1).<br>I see a lot of things posted on social media but unless I see a reference that I recognize on those like info charts and graphs or whatever I don't trust it, I will swipe right on my phone because it gives you a timeline of all the news articles and I'll skim over those, but I will also look for where they got their stats and information from (Spring 2020, Survey 1).<br>So in order to avoid a whole bunch of miscommunications I go straight to the website and I don't use other social media as my sources (Spring 2020, Survey 1).<br>If I see an article, I will probably look at where they were funded. Um, and looking at that, like, who gave them the money to do the research kind of thing, to judge if they are credible at this time (Spring 2020, Survey 1).<br>I try to go on extra ones just to try to verify stuff. And if most of them are reporting like the same thing and most of them are like reputable sources, then I'm more inclined to say, "okay" (Summer 2020, Survey 2).<br>Everywhere that I get it from, I always confirm it with a backup research. So, I always go to Google Scholar and like find something about it to confirm that it's right (Summer 2020, Survey 2).<br>If it's a blog post or a website, I actually look into who wrote the blog or what this website is about. And then I'll either close the article immediately, or I'll-I'll read it and take it with, like, a grain of salt (Summer 2020, Survey 2). |

(*Continued*)

**Table 10.** (Continued)

| Time | Actively searching for COVID-19 information | I've also been on the website of the World Health Organization and that's the one I refer to the most (Spring 2020, Survey 1).<br>Not too much like government sites until I started getting more paranoid and then I just like went to those to like check symptoms and stuff like that (Spring 2020, Survey 1).<br>I look at the, the WHO's situation report every day which gives a global overview of you know, which countries have new confirmed cases (Spring 2020, Survey 1). |
|---|---|---|
| | Passively taking in COVID-19 information | Listen to the news and that's, you know, only about Covid so in that instance it's passive for sure because it just happens to be about the virus (Spring 2020, Survey 1).<br>I can't avoid seeing it (Spring 2020, Survey 1).<br>Unless I'm like concerned about something specific, I usually just let it come to me just because I don't wanna think about it more than I have to at this point (Summer 2020, Survey 2).<br>If I listen to the radio, like while I'm driving, or if I see on my social media, like on Facebook, if I'm scrolling on Instagram, there's a post about how many cases there are. Then I'll, like, notice it. Otherwise, I'm not going and searching it up (Summer 2020, Survey 2).<br>I just kind of figured that if it's- if it's a big enough adjustment to the rule or whatever, somebody will let me know (Fall 2021, Survey 5).<br>Whenever it pops up I'll read it, but at this point I can't be bothered looking it up anymore (Fall 2021, Survey 5).<br>I'm already getting all the information without me searching for it (Fall 2021, Survey 5).<br>And I don't actively look for it, but it's always there (Spring 2022, Survey 6). |
| | Less than 'before' | So now I'm not, but before, I definitely was, like I was always looking. I wanted to know how many cases there were. I wanted to know like what countries are affected the most, but now, I'm kind of just passively, like if I hear about it's like, "Oh, okay" (Spring 2020, Survey 1).<br>I used to be checking it constantly 'cause I was like, "Oh, my God, I need to know everything." If there's something, update between now and two minutes ago, I need to know. Whereas like, now, I don't really care (Summer 2020, Survey 2).<br>I guess like the longer that it went, the less often I would check, 'cause it kind of the same- it was kinda the same information over and over again, I guess (Summer 2020, Survey 2).<br>Frankly, initially, I was actively looking for it. And I think after the second, third month of dealing with it, uh, I stopped looking um, not because I stopped caring, but because it became a reality on just an everyday basis, and I didn't want to consistently be reminded that I can't be going out because I know I can't go out (Summer 2020, Survey 2).<br>Because of midterms, I've not been researching a lot (Fall 2020, Survey 3).<br>Oh, before I was looking all the time. Like I would literally wake up and directly check the statistics if cases increase, decrease, but, to be completely honest, I haven't done so at all recently (Fall 2021, Survey 5) |
| Stress/Mood | Effect of overwhelming/negative media coverage | I feel like the more that I read about on a daily basis that OK this is the amount of people that passed away I feel like it's going to affect me like I'm just gonna start thinking about that (Spring 2020, Survey 1).<br>But I also try and limit the intake. Because I have like daily life and then, you know you don't want to like have your whole time be like coronavirus, like hearing bad news, good news, and let that sway your mood (Spring 2020, Survey 1).<br>I think after a while, I had to stop looking at the media and then I felt a bit- a little less overwhelmed. So that made me think OK, the more media I do consume, the more, the more um overwhelming it's seems in my head so, they definitely do go together (Spring 2020, Survey 1).<br>I mean, I think because it's so prevalent right now, that's like literally what everyone's talking about and if I'm gonna hear about, honestly, like depressing news every single second and then look for it as well, like that's not okay for my mental wellbeing (Summer 2020, Survey 2).<br>It's not that I don't recognize that COVID is-is still an imminent thing, it's just that constantly hearing about it has become kind of a burden (Fall 2020, Survey 3).<br>I think it is definitely just exhausting hearing and also worrying about COVID all the time (Fall 2020, Survey 3).<br>It's kind of a little draining emotionally, the- to see the same headlines over and over (Fall 2020, Survey 3).<br>But ever since I got my vaccine, which was two weeks ago-ish. I stopped looking. I'm exhausted (Spring 2021, Survey 4).<br>'Cause I think that also knowing and like updating yourself a lot is very dreary as well (Fall 2021, Survey 5). |

(*Continued*)

**Table 10.** (Continued)

| | | |
|---|---|---|
| *Family* | Family as source of COVID-19 information | It's basically, basically my parents because they seem to talk about nothing but that, but they kind of- they're telling me as they're reading it, so that's when I- when I'm there I kinda have to listen to it (Spring 2020, Survey 1). My dad just like sends me the Canadian Covid website link every once in a while, to be like, "look at it!" and I'm like, "cool, thanks dad" (Spring 2020, Survey 1). Mainly, I get it through mom or through Instagram 'cause this is where people are most likely to spend their time during quarantine and share the news (Spring 2020, Survey 1). I've been getting most of my information from my parents who are getting it online (Summer 2020, Survey 2). Like, my parents, they- my dad will sit there and sometimes I'll-I'll be like, "Oh, are you a news reporter," because he'll just be, like, saying the new-news out loud to me (Summer 2020, Survey 2). |
| | Keeping track of COVID-19 information for family | Especially these past two days, I have been researching a lot about vaccines. Um, because there's-there's a-a lot of misinformation going on regarding where we can book, um, COVID-19 vaccines for hotspots. I am- I am in one of the hotspots so I-I was researching where I can book, where I can't book, what's the procedure, um, and all that (Spring 2021, Survey 4). I was looking out mainly for like my mom- guys who were like in the 50, 45ish zone (Spring 2021, Survey 4). And I was, like all over about like 10, 12 different websites just. . . my mom, my dad, this and that sign up, sign up.. . . I clicked on this one website for my mom. . .And I jumped on it right away. . . I was like "here mom, here's the address, go." . . .Then my dad when I was signing him up, he was the next day, and I was like "do you have meetings at this time?" He's like "yeah. . ." I was like "okay, please figure something out so we can get this." Right? And again, like at that time, they were dishing out Astra Zeneca, I was like "doesn't matter!" I was like "doesn't matter–vaccine is vaccine, go!" (Spring 2021, Survey 4). |
| *Geographical focus* | Local | I think at the same time, like right now I'm kind of focused on just like, how is it impacting like the people around me or like my hometown, like kind of my communities that I'm a part of (Spring 2020, Survey 1). I think, uh, sometimes the news is just a bit too overwhelming, so I think just starting with Canada and even just with Toronto is like enough for me (Summer 2020, Survey 2). Right now I'm just mostly focusing on, like, what the regulations are and that's it (Summer 2020, Survey 2). It's just a bit overwhelming, just, uh, having all that information. So just staying local is good enough for me for now (Fall 2020, Survey 3). I'm mainly focusing on local news now. I've, uh, stopped looking internationally because I-I feel like-like the closest to my house is probably like the local, I don't-I don't think international is gonna really affect us (Spring 2022, Survey 6). |
| | Provincial | I wouldn't care about what they're doing in Alberta right now, I-I wanna know what they're—what's happening in Ontario (Summer 2020, Survey 2). So especially like- I'm an international student, so I have to travel home. So around the time of like when I'm traveling home I check like Ontario case numbers because I'm like "I hope they're not too high, because I have to get a COVID test to go home" or like when I'm coming back to Canada I also check that kind of thing (Fall 2021, Survey 5). Like obviously in my situation trying to learn about the vaccine and all those things sort of sparked my interest in provincial news (Spring 2021, Survey 4). |
| | National (Canada) | I kind of try to—I keep the focus on Canada to kind of keep—also feel kind of contained in it if that makes sense (Summer 2020, Survey 2). |
| | International (where family/friends are) | I'm also very, very in tune to the United States situation right now because I lived in the States for 10 years and I also have family and loved ones there (Spring 2020, Survey 2). I'm from Hong Kong, so I have, um, I stayed, keep in touch with regarding to Hong Kong COVID-19 as well (Summer 2020, Survey 2). I do look for the international picture, although I think that's strongly swayed by the fact that I have family abroad (Summer 2020, Survey 2). |
| | International (for case count/ mandate comparisons) | But I try and get a more, more of a global perspective to see how other- like if other countries are doing something, what they're doing about it, what the consequences are (Spring 2020, Survey 1). I'm more likely to- press on a button to show me what Italy's currently dealing with-than I am most other countries, uh, just because, um, I feel they've been hit by this quite strongly, and if somehow Italy can make it out of this, and hopefully they do, um-then it's-it's quite a positive thing for the rest of the world as well as a morale booster (Summer 2020, Survey 2). I find it interesting to see, like, the rankings and, like, to see, like, the second waves go through different places that were hit first. Um, and to, like, basically maybe gauge how that ours will become. Because, like, you'll see places like France and Italy and Spain who were, like, hit very early on and now they're going through a second wave and they're getting like 20,000 cases and it's like, "Oh, wow, [laughs] are we headed there if we don't, like, start shutting things down again?" (Summer 2020, Survey 2). |

seeking information regarding changes or updates to the COVID-19-related rules and mandates (e.g., masking on public transit, closure/opening of the American-Canadian border).

*Passively taking in COVID-19 information*. Interviewees consistently expressed that they were passively taking in COVID-19 information throughout all sets of interviews. This included walking by a television their family was watching, hearing information on the radio in their car, seeing news headlines appear on their smartphone push notifications, and scrolling past headlines on various forms of social media.

*Less than 'before'*. Starting in Summer 2020 (Survey 2), students expressed that they had lessened their intake of COVID-19 information since either the last time they were interviewed or, for the new participants, since earlier in the pandemic. In this survey one participant described their shift in media consumption: "*I used to be checking it constantly 'cause I was like,* "*Oh, my God, I need to know everything.*" *If there's something, update between now and two minutes ago, I need to know. Whereas like, now, I don't really care.*" An exception appeared in Spring 2021 (Survey 4), when students noted that they were spending more time looking for information on where to book vaccines.

**Stress/mood.** *Effect of overwhelming/negative media coverage*. In the first five rounds of interview, students highlighted the effect of COVID-19-related media coverage on their levels of stress and noted it negatively affected their mood or made them feel overwhelmed. In Survey 1 one participant noted that "*I have like daily life and then, you know you don't want to like have your whole time be like coronavirus, like hearing bad news, good news, and let that sway your mood*" while a participant in Survey 5 explained that "*it's just that constantly hearing about it has become kind of a burden.*"

**Family.** *Family as source of COVID-19 information*. In the Spring and Summer 2020 interviews (Surveys 1 and 2), students noted that their families (either in person during lockdown, or via communication channels such as WhatsApp and WeChat) were key sources of information regarding COVID-19, though not always welcome ones. This theme was absent in Fall 2020 (Survey 3) and Spring 2021 (Survey 4) but appeared again in Fall 2021 (Survey 5) and Spring 2022 (Survey 6) as students noted that they were no longer actively seeking information, but mostly passively getting information from their families.

*Keeping track of COVID-19 information for family*. In the Spring 2021 (Survey 4) interviews, interviewees noted their role in finding information regarding vaccine appointments for their families. This was the only interview in which this topic emerged.

## Geographic focus

The geographic focus of the information accessed by the interviewees narrowed over time as interview participants indicated they were spending less time overall engaging with information related to COVID-19. International news for case count and mandate comparisons was common in the three 2020 interviews but was not mentioned from Spring 2021 (Survey 4) onwards. International news in locations where family and friends of the participants live was commonly discussed from Spring 2020 (Survey 1) to Spring 2021 (Survey 4) but was limited only to information regarding travel restrictions to those locations by Fall 2021 (Survey 5). Similarly, national news for Canada was a focus from Spring 2020 to Spring 2021, but in Fall 2021 was only mentioned in relation to travel restrictions and in Spring 2022 (Survey 6) in relation to national levels of vaccine uptake. Provincial news (i.e., Ontario or location of student's residence) was mentioned in the first five rounds of interview, but not in Spring 2022. Local news, relating to the participants' city and/or health region of residence was consistently highlighted in all rounds of interview. Table 10 also indicates topics newly introduced by the

interviewees as notable in their media consumption and those topics which were repeatedly mentioned in later interview iterations.

## Discussion

This research investigated the demographic factors driving perceptions of severity, susceptibility, and health behaviour adoption over time, considering media sources as a cue to action within the Health Belief Model. The findings are represented in Fig 6, adapted from Rosenstock's visualization [11]. This study found that female gender was a driver in judgements of severity, susceptibility, and the adoption of new health behaviours over time. These results align with other studies concerning COVID-19, that have generally found that female gender is associated with higher risk perception, higher perceptions of vulnerability or susceptibility, and adoption of preventative health behaviours in both cross-sectional investigations [3, 26–30] and those targeting university-age students globally [5, 31–35], although this is not a universal finding [36].

Female gender driving higher perceptions of disease severity has also been found in both international and localized studies regarding previous outbreaks such as SARS [37] and Zika [38]. Survey research such as these provide critical snapshots during developing health events that can assist with targeting health communication strategies towards specific demographic groups. Longitudinal surveys, however, are necessary for capturing changing perceptions during rapidly changing health crises. For instance, in our previous work concerning Survey 1 [6], female university students indicated a higher perceived severity and susceptibility, but these results were not found to be statistically significant; however, when Surveys 1 to 6 were examined together in the present study, female gender emerged overall as a significant factor in both categories.

Overall, perceptions of severity fell steadily from Survey 1 to Survey 6. A similar result was found by Bults et al. [39] using three consecutive surveys of the general public in the Netherlands during the 2009 swine flu outbreak, in which the perceived severity of swine flu dropped between April and August. In contrast, de Zwart and colleagues [40] conducted a one-year survey (2006–2007) of adults in the Netherlands to track perceptions of avian influenza and found that perceived severity was high throughout seven consecutive surveys. These mixed results emphasize the importance of consecutive surveys to track changes over time and investigate what modifying factors might be driving them.

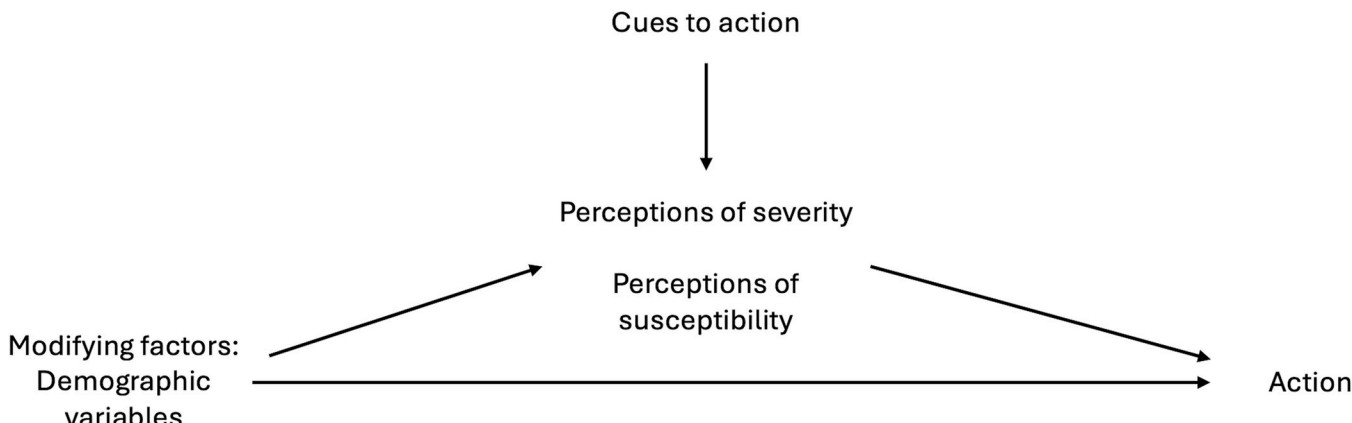

**Fig 6. Results of the study interpreted within the Health Belief Model (adapted from Rosenstock [13]).**

Age was found to be significantly related to perceptions of susceptibility; with every one-year increase in age, perceptions of susceptibility rose by 17%. Studies such as Rosi et al. [41] and de Bruin [42], surveying samples including young adults to elderly individuals, found that perceived risk vulnerability to COVID-19 decreased with age. Others, such as Niño and colleagues [27] found older adults reported higher risk perceptions of COVID-19. These mixed results demonstrate the importance of fine-grained analysis of smaller age cohorts, as the nuanced differences in the present study reveal.

Judgments of susceptibility changed over time; perceptions increased between Surveys 1 and 3, then dropped to the lowest levels in Survey 5 (Fall 2021, pre-Omicron) before increasing to the highest levels in Survey 6 (Spring 2022). The consecutive surveys capture these fluctuations in perceptions, as case counts rose in Canada (and internationally), in the interviews more participants indicated they either they had contracted COVID-19 or knew someone who had. The emergence of the Omicron variant in November 2021 is a possible driver of the high perceptions of susceptibility by the final survey in Spring 2022. The declining perception of severity concurrently with varying perceptions of susceptibility is an important finding, as public health communication efforts may not have been reaching university students, as their overall media intake decreased over time, as discussed below.

The relationship of new health behaviours with female gender has been reported in other survey studies regarding COVID-19 [3, 43] and previously with regards to outbreaks such as SARS [44, 45] and swine flu [46, 47]. This is not universal when considering specific health behaviours, however, as studies regarding face mask use during SARS did not find a relationship between mask wearing and gender [48, 49]. González-Castro and colleagues [50] conducted surveys of Spanish adults in March and April 2020 and found that the adoption of health behaviours increased, demonstrating the value of consecutive surveys in the same location. The patterning of health behaviours over time in the present study likely reflects the effects of health behaviour mandates over time. Between Survey 1 and Survey 2, COVID-19 case counts increased rapidly and the population of Ontario was encouraged to wear masks [51]. Between Surveys 2 and 3, mask mandates were introduced across the province in healthcare settings and, during the Survey 3 data collection period, a provincial mandate was introduced for indoor public areas and transit [52]. These longitudinal results capture the influence of provincial mandating of mask wearing, the introduction of vaccine mandates for university students, the introduction and removal of travel restrictions (e.g., closure of USA-Canada border to non-essential travel, vaccine requirements for air travel), and vaccine passports to access services such as indoor dining [53–56]. Such cues to action are critical to consider within the Health Belief Model, as it is impossible to separate the self-reported health behaviours from the influence of government mandates. The highest level of reported health behaviours appeared in Survey 3 (Fall 2020) when mask mandates were in place, university classes were nearly completely offered online, and in-person gatherings were limited. Notably, non-mandated behaviours such as increased cleaning, buying extra supplies, and hand washing, declined steadily throughout the survey periods.

Participants' judgement of feeling fearful/anxious after consuming information about COVID-19 was associated with female gender and low income, and decreased steadily over the research period. Our results resonate with broader research concerning the negative mental health effects found in female university students globally in the early pandemic period [57, 58]. Regarding income, Gill et al. [59], surveying a group of Canadian young adults (18–24 years), found a relationship between individuals with reduced income due to the pandemic who were receiving government aid and symptoms of psychological distress regarding COVID-19. Park and colleagues [60], in a population-based survey of Americans in the early pandemic period, reported that finances were rated the most stressful aspect of the pandemic.

Similar results were reported by Fischhoff and colleagues [61] regarding the American public and Ebola, in which female individuals and those with lower incomes reported higher worry. Worries regarding financial strain (e.g., lost employment for self or family member(s), few employment opportunities, tuition and rent payments) were likely driving this relationship. A meta-analysis conducted by Strasser and colleagues [62] found that the increase of pandemic-related information consumed on social media was associated with declining mental health in young adults, thus the decline in COVID-19 information intake as a self-protective strategy is possible.

Research regarding disaster exposure (e.g., natural disasters, pandemics) and media has indicated a relationship between media use and adverse mental health reactions such as stress and depression [63–67]. Interestingly, we did not find a significant association between the amount of media time reported by participants and judgements of anxiety/fear in this analysis. The interview participants, however, did express that the media coverage of COVID-19 heightened feelings of stress and negatively impacted their mood. Perceptions expressed by interview participants included that fact that COVID-19-related news was excessive, which may have repercussions including social media fatigue leading to information avoidance behaviours [66].

Participant engagement and perceptions of media use for COVID-19 information were investigated. Overall, the time spent using all forms of media decreased over the survey period. This result was supported by the interviews, in which interviewees in each round indicated that they were looking at media less, citing feeling overwhelmed or exhausted by the media coverage. Following Antunovic et al.'s [68] three-stage process of news consumption, the interview participants indicated that their routine surveillance (e.g., scrolling through phone news apps in the morning) and directed consumption (i.e., following a headline to the full article, looking up information) decreased over the research period. The third stage, incidental exposure, continued throughout the research period, for example in the form of family talking about COVID-19 over dinner or overhearing the TV news that a family member was watching in the participant's house. These incidental exposures initially led to directed consumption, but by Survey 4 participants made it clear they actively sought only specific information regarding vaccine availability and changing government mandates. The theme of family in reference to COVID-19 information was key, as participants described the role of their family in providing COVID-19 information and seeking information for their families in reference to booking family members' vaccine appointments.

Identification of the sources of knowledge on disease provides insight into which information sources appeal to and are considered authoritative by university students. In this study, social media was identified by participants as their most used information source, similar to other international studies querying students' media use regarding COVID-19 [8, 69]. At the same time, students indicated that social media was most likely to cause feelings of fear and anxiety about becoming infected with COVID-19 and over time participants judged social media's reliability for accessing clear and unbiased information regarding COVID-19 to move from middling to poor. Interview participants highlighted sources such as Canadian news outlets and websites, the World Health Organization, and the Centres for Disease Control as reputable sources. In contrast, social media was characterized as biased and untrustworthy.

Similar results were found in a contemporary survey (March 24–27, 2020) to our Survey 1 of medical students in Turkey, 50.8% of whom used social media to find COVID-19-related information, but 82.0% of whom stated they did not trust information found through social media [70]. University students in Palestine surveyed in April 2020 also indicated that they trusted social media the least as a source of information regarding COVID-19 [71] and a survey of primary and secondary students (ages 6–18) from Palestine found a significant

relationship between panic about COVID-19 and the use of social media [72]. This is a key finding as public health professionals and university health centres globally seek to harness the power of social media to disseminate COVID-19-specific information [73–75] as digital health literacy among university students has been found to be uneven [76].

In Survey 1 and 2, interview participants offered descriptions of strategies they used for ascertaining the reliability of COVID-19 news sources. By Survey 3 (Fall 2020), interview participants were no longer describing such strategies, as their COVID-19 media time decreased overall. Participants described actively searching for information in all three 2020 surveys, but only in reference to specific rules surrounding vaccine information and government mandates by Survey 4 (Spring 2021). Concomitantly, the geographic focus of the information gathering decreased through time, as interviewees expressed the need only to look up information that affected them in their quotidian activities, rather than spending significant time studying the national or international situation. Within the Health Belief Model, this represents individuals' beliefs affecting their perception of how they engage with a cue to action (i.e., decreased media use overall and decreased geographical range) (represented in Fig 6). In the earlier surveys, participants described the desire for an international perspective and to estimate how waves present in other countries could affect Canada in the future. This type of investigation disappeared from the interviews by Survey 4 (Spring 2021), where international information was only sought for travel restriction rules updates. Local information remained the priority as participants posited that international information was not relevant.

Survey participants also indicated the use of television and internet news media for COVID-19 information. This resonates with Antunovic et al. [68], who found that young adults use a range of media sources for news gathering, and generally have greater trust in traditional news media [68, 77]. In this study, the majority of participants consistently judged internet news, television, radio, and magazines to be of "middling" quality in providing clear, concise, and unbiased information regarding COVID-19. Radio and magazines were used infrequently as information sources, with radio being used more often by commuters who drove their own vehicles. This result is worthy of further investigation, since previous nationally representative research of Americans concerning Zika media coverage found that while the volume of social media coverage influenced risk perceptions, the legacy media (television and newspapers) coverage correlated with health protective behaviours [78]. While they were not used as frequently, these legacy media (and internet news sources) could be a better platform for reaching university students in future disease events.

Media use represents a cue-to-action for young adults within the Health Belief Model, where their own pre-existing beliefs can be altered through exposure to an outside source. Individual perceptions of severity and susceptibility were clearly influenced by media consumption, as those who felt anxiety after reading media reports about COVID-19 were 8.43x more likely to rate the severity of COVID-19 as high, and 2.07x more likely to indicate their own susceptibility was high. Participants combine media information with their own existing knowledge on COVID-19, leading to fluctuations in perceptions of severity and personal susceptibility. Overall, as the media use time decreased through the survey period, so did the severity ranking. Beliefs about susceptibility were more complex and involved a greater reliance on modifying factors (e.g., mask wearing, age, travel) (see also Mant et al. [6]). It is notable that the severity rating decreased steadily over the study period, while the average susceptibility rating was more varied at each survey, potentially in reference to the context of government mandates and the emergence of new variants.

These results contribute to an understanding of university students' information-seeking habits and perceptions during a global pandemic and further, how media use, self-declared feelings of anxiety/fear, and perceptions of severity and susceptibility are interrelated. Media

research has indicated that information-seeking habits formed "in the late teens and early 20s as individuals enter the world of independent adulthood" can become habituated and "routinized" [68, 79, 80]. This routinization has important implications for the ongoing pandemic and future outbreaks, in which university students will likely have routinized their media consumption patterns. If the media source that university students use the most is also the one they trust the least, it is necessary for public health communication to consider this complex relationship when creating future public health campaigns. This lesson is important for communication beyond pandemic issues. Consistent, real-time, clearly cited messaging with localized resources is necessary to engender trust and encourage action. As noted above, young adults are often subsumed into 'adult' (or 18+ years) categories, which collapses the unique concerns of this group. Public health communicators should consider partnerships with universities and colleges to both build trust in the health information and maximize information dissemination (i.e., internal listservs).

## Strengths, limitations, and future research

The findings of this study are based upon a convenience sample at one university and may not be generalizable to other university student groups or broader age or international contexts. Web-based surveys demand self-selection, thus our study sample is made up only of students who saw the survey link and self-selected to complete the survey. Participants were not linked between survey iterations thus the quantitative contribution cannot be analyzed as a cohort study. To address this limitation, repeat interviewees were encouraged to reflect upon how their perceptions had changed over time (S1 Text). The study also does not establish students' baseline anxiety level or use clinically validated measures of depression, anxiety, or stress symptoms. Survey participants were asked to self-report their media usage. Asking participants to report their recent consumption of COVID-19-related news in each survey rather than estimating past consumption may mitigate some of this potential source of bias. Despite these limitations, this study is the first to our knowledge to have surveyed Canadian university students regarding their perceptions of the COVID-19 pandemic, and contributes important longitudinal data on rapidly changing attitudes towards the ongoing pandemic. More mixed-methods research should be conducted to investigate variability in university student responses across Canada and globally. Important insights regarding lessons learned from COVID-19 and health behaviours long term could be gained by interviewing a cohort of the interviewees of this research.

## Conclusion

As the COVID-19 pandemic continues, public health communication regarding booster vaccinations, viral mutations, mandates, rules, cases, and hospitalizations will continue to be relevant. Perhaps more importantly, the overall increased perception of individual susceptibility to COVID-19 over the first two years of the pandemic alongside decreased perceptions of severity is a key result. Longitudinal monitoring is important to understanding an ongoing disease event, evidenced by the fact that time of survey was significantly associated with perceptions of severity, susceptibility, 10 recorded health behaviours, and judgements of anxiety/fear after interacting with a COVID-19 related media report. Public health officials and universities seeking to communicate health behaviour information to university students should be aware of the fatigue expressed by students regarding such information as the pandemic continues; this is relevant to future disease communication strategies for this and future pandemics. Avoidance of media due to feelings of being overwhelmed or stressed are critical to note, particularly in ongoing disease events. Public health leaders must consider the appropriate

junctures to introduce new and/or updated messaging to ensure the message is received and potentially cues university students to action.

## Supporting information

**S1 Text. Survey and interview guides: Survey Guide Round 1: March/April 2020; Survey Guide Round 2: June/July 2020; Survey Guide Round 3: September/October 2020; Survey Guide Round 4: March/April 2021; Survey Guide Round 5: September/October 2021; Survey Guide Round 6: March/April 2022; Interview Questions.**
(DOCX)

**S2 Text. Additional results tables: Table A.** Likelihood ratio tests of demographic predictors for judgement of severity of COVID-19, susceptibility to COVID-19, and adoption of new health behaviours; Table B. Binary logistic regression results for judgement of severity of COVID-19, susceptibility to COVID-19, Poisson regression results for new health behaviours; Table C. Likelihood ratio tests of demographic predictors for feelings of anxiety/fear after hearing/reading a COVID-19 news report; Table D. Logistic regression analysis results for anxiety/fear after hearing/reading a COVID-19 news report; Table E. Multinomial logistic regression results regarding the forms of media that made participants feel most anxiety/fear about becoming infected with COVID-19; Table F. Binary logistic regression for judgement of severity of COVID-19; Table G. Interview participants' demographics.
(DOCX)

## Author Contributions

**Conceptualization:** Madeleine Mant, Andrew Prine, Alyson Holland.

**Data curation:** Madeleine Mant, Alyson Holland.

**Formal analysis:** Madeleine Mant, Asal Aslemand, Andrew Prine, Alyson Holland.

**Funding acquisition:** Madeleine Mant, Andrew Prine, Alyson Holland.

**Investigation:** Madeleine Mant, Andrew Prine, Alyson Holland.

**Methodology:** Madeleine Mant, Asal Aslemand, Andrew Prine, Alyson Holland.

**Project administration:** Madeleine Mant.

**Resources:** Madeleine Mant.

**Supervision:** Madeleine Mant.

**Validation:** Asal Aslemand.

**Visualization:** Asal Aslemand.

**Writing – original draft:** Madeleine Mant, Asal Aslemand, Andrew Prine, Alyson Holland.

**Writing – review & editing:** Madeleine Mant, Asal Aslemand, Andrew Prine, Alyson Holland.

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
