## [Decision Letter · Decision Letter 0]

4 Apr 2024

PGPH-D-24-00108

Mixed-methods study of university students’ perceptions of COVID-19 and media consumption from March 2020 – April 2022

Dear Dr. Mant,

Thank you for submitting your manuscript to PLOS Global Public Health. After careful consideration, we feel that it has merit but does not fully meet PLOS Global Public Health’s publication criteria as it currently stands. Therefore, we invite you to submit a revised version of the manuscript that addresses the points raised during the review process.

Editor comments:

The reviewers and I found the manuscript to be well-written, and feel that it will make a useful contribution to the growing literature on young adult experiences in the context of COVID-19. However, there are several places where additional methodological clarity would improve the manuscript.Specifically, both reviewers and I have several questions regarding the methods and design. The approach used to integrate qualitative and quantitative findings needs further description. In addition, I was unclear as to why the qualitative results were only partially presented in the body of the results, with the bulk of the material included in a supplemental file. Given that the journal does not have a specific word count, and it make it harder on the reader, I would encourage the authors to include the qualitative findings in the main manuscript. If there are additional quotes or secondary materials, then these could be placed in a supplemental file. Further, qualitative quotes should not be presented in the Discussion. They should be presented and analyzed in the results, with the other data. 

We look forward to receiving your revised manuscript.

Kind regards,

Marie A. Brault, PhD

Academic Editor

Journal Requirements:

Additional Editor Comments (if provided):

Reviewers' comments:

Reviewer's Responses to Questions

**Comments to the Author**

1. Does this manuscript meet PLOS Global Public Health’s publication criteria? Is the manuscript technically sound, and do the data support the conclusions? The manuscript must describe methodologically and ethically rigorous research with conclusions that are appropriately drawn based on the data presented.

Reviewer #1: Yes

Reviewer #2: Yes

2. Has the statistical analysis been performed appropriately and rigorously?

Reviewer #1: Yes

Reviewer #2: Yes

3. Have the authors made all data underlying the findings in their manuscript fully available (please refer to the Data Availability Statement at the start of the manuscript PDF file)?

Reviewer #1: Yes

Reviewer #2: No

4. Is the manuscript presented in an intelligible fashion and written in standard English?

Reviewer #1: Yes

Reviewer #2: Yes

5. Review Comments to the Author

Reviewer #1: The aim of the manuscript is to investigate the perceptions and behaviors of university students regarding the COVID-19 pandemic over a period of time through a mixed method approach. The study aims to explore how various factors, including gender, age, media consumption, and government mandates, influence students’ perceptions of severity, susceptibility, adoption of health behaviors, and feeling of anxiety/fear. It also provides insights into the sources of COVID-19 information preferred by university students and how these sources impact their perceptions and behaviors.

INTRODUCTION:

The paper suggests using a mixed-methods approach for understanding changing perceptions and behaviors. It could be clearer on why both quantitative and qualitative data are needed and how they work together to tackle the research questions.

METHODS:

Design:

The paper describes the mixed methods design used in the study, which appears to be concurrent, with qualitative and quantitative data collected simultaneously. A visual display of the overall design highlighting integration, which could enhance clarity for readers.

Subjects/Participants:

The sampling and recruitment is adequately described. Please give an estimation of the response rate.

Are there any anonymous identifiers identifying participants who answered multiple rounds of the surveys.

Methodological Rigor:

There is no discussion of efforts to ensure data quality, such as data cleaning procedures or inter-rater reliability checks, member checking, triangulation, or flexibility. Without mentioning these aspects, it’s challenging to assess the rigor of the quantitative methods employed.

RESULTS:

All components of the study findings have been clearly presented, including both qualitative and quantitative results. The integration of findings from both methods has been effectively demonstrated, such as through joint displays.

DISCUSSION:

This section effectively incorporates the implications of the integrated findings, provides synthesis and interpretation of the results in the context of existing literature and theoretical/conceptual frameworks, and includes a subsection on limitations.

Strengths:

Effectively synthesizes the findings, drawing connections between the results and existing literature.

The integration of qualitative interview data enriches the discussion, providing insights into participants’ experiences and perceptions.

The discussion acknowledges the limitations of the study, such as the convenience sample and the potential lack of generalizability.

Suggestions for Improvement:

Explore potential practical applications or recommendations

Providing links between the study’s theoretical/conceptual framework and the interpretation of findings could strengthen the theoretical grounding of the discussion.

Consider discussing potential avenues for future research or areas where further investigation is needed to build upon the current findings.

Reviewer #2: Dear Editor,

Thank you for providing the opportunity to review this work. The authors have conducted a study to describe the changes in risk perception and factors associated with it. They also describe changes in preventive practices over time throughout the study, as well as the influence of exposure to communication media. The manuscript is very well written and clearly divided into sections and subsections, which facilitate easy reading despite the lengthy list of results presented by the authors. I have no major comments; however, there are some minor points that need clarification or improvement before publication.

Study design: The authors need to describe the study design clearly. They mention that they conducted 6 surveys; however, it is not clear whether there are participants who could have participated 2 or more times. In such a case, it could be a cohort study requiring adjustment for repeated measures by time and individual. Alternatively, if they were unable to confirm the identity of all participants, this should be addressed as a limitation. Lastly, if it was ultimately a serial cross-sectional study, this should be clarified.

Population and sample: The authors indicate that the sample was convenience-based. It would be possible to perform a sensitivity analysis using age, sex, faculty, and other variables with which we can evaluate the university in general versus the sample to determine how representative the sample is, thereby understanding the effect of this. Perhaps it would be interesting to conduct a post-stratification analysis.

External validity: In addition to the convenience sample, the authors conducted the study with a specific population of university students. This reduces the external validity of their results and should be further detailed when making comparisons with other studies and in the limitations section.

Additional figure: It would be interesting to include a figure depicting the curve of cases and deaths associated with the region or country, with added lines or areas delineating the collection periods. Also, indicating important periods of the pandemic such as lockdown policies or mandatory mask usage would provide context for the pandemic in the study area.

Finally, as a suggestion, it would be interesting to see Directed Acyclic Graphs (DAGs) for different outcomes and interrelated factors. It would also be interesting to see a unified DAG following the theoretical framework that the authors employed for the study. Additionally, discussing whether there is a way to unify all the results and factors would be beneficial to identify single points of public health intervention that can influence multiple outcomes.

6. PLOS authors have the option to publish the peer review history of their article (what does this mean?). If published, this will include your full peer review and any attached files.

**Do you want your identity to be public for this peer review?** For information about this choice, including consent withdrawal, please see our Privacy Policy.

Reviewer #1: No

Reviewer #2: **Yes: **Juan P. Aguilar Ticona

---

## [Decision Letter · Decision Letter 1]

9 Jun 2024

PGPH-D-24-00108R1

Mixed-methods study of university students’ perceptions of COVID-19 and media consumption from March 2020 – April 2022

Dear Dr. Mant,

Thank you for submitting your manuscript to PLOS Global Public Health. After careful consideration, we feel that it has merit but does not fully meet PLOS Global Public Health’s publication criteria as it currently stands. Therefore, we invite you to submit a revised version of the manuscript that addresses the points raised during the review process.

Editor comments:

The reviewers and I appreciate the authors' responsive to the previous comments, and feel that the manuscript is nearly ready to be accepted. However, there are a few areas requiring revision before this can be accepted (as no edits other than copy-editing are allowed once the manuscript is accepted).Please address Reviewer 2's questions concerning the selection model, and comments regarding Table and DAG formatting for consistency and clarity.

We look forward to receiving your revised manuscript.

Kind regards,

Marie A. Brault, PhD

Academic Editor

Journal Requirements:

b. If any authors received a salary from any of your funders, please state which authors and which funders.

Additional Editor Comments (if provided):

Reviewers' comments:

Reviewer's Responses to Questions

**Comments to the Author**

1. If the authors have adequately addressed your comments raised in a previous round of review and you feel that this manuscript is now acceptable for publication, you may indicate that here to bypass the “Comments to the Author” section, enter your conflict of interest statement in the “Confidential to Editor” section, and submit your "Accept" recommendation.

Reviewer #1: All comments have been addressed

Reviewer #2: All comments have been addressed

2. Does this manuscript meet PLOS Global Public Health’s publication criteria? Is the manuscript technically sound, and do the data support the conclusions? The manuscript must describe methodologically and ethically rigorous research with conclusions that are appropriately drawn based on the data presented.

Reviewer #1: Yes

Reviewer #2: Yes

3. Has the statistical analysis been performed appropriately and rigorously?

Reviewer #1: Yes

Reviewer #2: No

4. Have the authors made all data underlying the findings in their manuscript fully available (please refer to the Data Availability Statement at the start of the manuscript PDF file)?

Reviewer #1: Yes

Reviewer #2: Yes

5. Is the manuscript presented in an intelligible fashion and written in standard English?

Reviewer #1: Yes

Reviewer #2: Yes

6. Review Comments to the Author

Reviewer #1: (No Response)

Reviewer #2: Thank you for the opportunity to review this work again. The authors have provided additional information and details that significantly enhance their work. They have clearly addressed all my questions and incorporated the necessary information into the text. I have aditional suggestions to further improve their work:

1. In the methods section, include a description of the selection model and how the performance between the models was evaluated. Justify the inclusion of the variables in the model and explain why sociodemographic factors were not included in the binary logistic regression for assessing the severity of COVID-19. If a strategy was used for model selection, it is important to evaluate and show the changes in the coefficients between the saturated model and the final model.

2. In Table 3, use consistent labels for the surveys (e.g., Survey 1, Survey 2, ... Survey 6) rather than seasonal terms like Fall and Spring. Standardize the terms in the manuscript to avoid misunderstandings.

3. Ensure that Tables 3, 4, and 7 use the same format, including the categories the frequency and percentage, to clearly show the direction of changes over time and the differences.

4. For the DAG figures, use the following reference to standardize the symbols: Digitale JC, Martin JN, Glymour MM. Tutorial on directed acyclic graphs. J Clin Epidemiol. 2022 Feb;142:264-267. doi: 10.1016/j.jclinepi.2021.08.001. Epub 2021 Aug 8. PMID: 34371103; PMCID: PMC8821727.

7. PLOS authors have the option to publish the peer review history of their article (what does this mean?). If published, this will include your full peer review and any attached files.

**Do you want your identity to be public for this peer review?** For information about this choice, including consent withdrawal, please see our Privacy Policy.

Reviewer #1: No

Reviewer #2: No

---

## [Editor Report · Decision Letter 2]

2 Jul 2024

Mixed-methods study of university students’ perceptions of COVID-19 and media consumption from March 2020 – April 2022

PGPH-D-24-00108R2

Dear Dr. Mant,

We are pleased to inform you that your manuscript 'Mixed-methods study of university students’ perceptions of COVID-19 and media consumption from March 2020 – April 2022' has been provisionally accepted for publication in PLOS Global Public Health.

Best regards,

Marie A. Brault, PhD

Academic Editor